# Assessing the effectiveness of no-take zones on fish populations in the Marine Natural Park of Cap Corse and Agriate, Northwestern Mediterranean Sea

Lucie Vanalderweireldt[1,2,3,4]*, Robin Bauknecht[1], Jessica Garcia[3,4,5], Manon Fournier[3,4], Christelle Paillon[3,4,6], Antoine Brias[3,4], Nicolas Tomasi[7], Anthony Caro[8], Eric D. H. Durieux[3,4]*

**1** Ecology and Landscape Evolution Unit, ETH Zürich, Zürich, Switzerland, **2** Land Change Science Unit, Swiss Federal Research Institute WSL, Birmensdorf, Switzerland, **3** Sciences Pour l'Environnement Unit, Université de Corse Pasquale Paoli, Corte, France, **4** Stella Mare Unit, Université de Corse Pasquale Paoli, Corte, France, **5** Biodiversité Halieutique Unit, IFREMER Cayenne, Guyane, France, **6** Station Marine MNHN, Muséum National d'Histoire Naturelle, Dinard, France, **7** Office Français de la Biodiversité, Marine Natural Park of Cap Corse and Agriate, France, **8** Office Français de la Biodiversité, Marseille, France

* lucie.vanalderweireldt@gmail.com; durieux@univ-corse.fr

## Abstract

No-take zones (NTZs) are expected to rebuild exploited fish populations, yet their performance is rarely assessed with species-level indicators. We quantified the reserve effect of the *Nonza–San Fiurenzu* NTZ (24.2 km$^2$) inside the Marine Natural Park of Cap Corse and Agriate (north-western Mediterranean) and simultaneously compared the effectiveness of two underwater visual census (UVC). Using 488 underwater visual-census transects (2018–2022) we monitored densities, size structure and biomass of three emblematic predators—the common dentex *Dentex dentex*, dusky grouper *Epinephelus marginatus* and brown meagre *Sciaena umbra*—and compared trends with neighbouring fished sectors. Gaussian GLMs related log-transformed density and biomass to protection status, season, year, sea-surface temperature (SST) and chlorophyll-*a*. Protection was the dominant predictor for *E. marginatus* and *S. umbra*: mean densities inside the NTZ were 3–4 times higher than outside, and biomass was enriched by factors of six and four, respectively. For the mobile *D. dentex* densities did not differ, but biomass was greater in the NTZ, indicating size-selective benefits. SST showed a negative effect on *D. dentex* and *S. umbra*; chlorophyll-*a* had no detectable influence. Cross-transect surveys (two divers) yielded density estimates comparable to—or higher than—the logistically intensive comb protocol (four to eight divers), suggesting that simpler designs can suffice for long-term monitoring. Our results demonstrate that a well-enforced NTZ embedded in a moderately protected park can rapidly enhance biomass and restore size structure of vulnerable Mediterranean predators.

**Data availability statement:** All data and code used in this analysis, including those required to

reproduce the results and figures presented in the paper, are publicly available at: https://doi.org/10.5281/zenodo.15846816.

**Funding:** This research was funded by the Agence de l'Eau Rhône Méditerranée Corse through convention No. AERMC-2017-1420 (recipient: E.D.H. Durieux) as part of the MoPaMFish project (Monitoring of Patrimonial Mediterranean Fishes) coordinated by Università di Corsica Pasquale Paoli (UCPP). Additional funding was provided by the French Inter-Regional Direction of the Mediterranean Sea (DIRM) under Convention No. 2021-DIRM-MICO-01 (recipient: N. Tomasi) to support monitoring activities within the Natural Marine Park (Parc Naturel Marin), with further contributions from the French Ministry of the Sea. The study was also part of the MoonFish project (Modelling tools for sustainable management of fisheries resources in Corsica), funded by the Corsica Region (Collectivité Territoriale de Corse, CdC) under the European PO-FEDER 2014–2021 funding framework (recipient: E.D.H. Durieux). MoonFish involved collaboration between the University of Corsica/CNRS (project lead), the Environment Office of Corsica (OEC), STARESO, and CRPMEM.

**Competing interests:** No authors have competing interests.

## Introduction

The Mediterranean Sea, a semi-enclosed oligotrophic basin, is renowned for its diverse ecosystems and rich biodiversity, hosting around 6.3% of the world's marine biodiversity, including 10.8% of endemic species [1]. However, these precious marine ecosystems have suffered detrimental impacts from a blend of human-induced ecological shifts, including climatic factors such as rising temperatures, and acidification, alongside pressures from fishing, maritime traffic, terrestrial pollution, and invasive species [2–4]. Thus, the Mediterranean Sea finds itself at the epicenter of cumulative threats, where intense human pressures overlap with its exceptional biodiversity, placing the region effectively under siege [4,5].

As a biodiversity hotspot, the Mediterranean Sea hosts numerous patrimonial and threatened species, including the common dentex (*Dentex dentex*), the dusky grouper (*Epinephelus marginatus*), and the brown meagre (*Sciaena umbra*) respectively listed as vulnerable, endangered, and near threatened for the last one (IUCN Red List) [7–9]. In this semi-enclosed basin, accelerated warming and climate-driven trophic shifts [2,4,6] interact with fishing pressure to further reduce predator diversity, favoring smaller-bodied fishes. Besides, these three species are further susceptible due to their large size, slow growth, and low reproductive rates [10,11], making them especially sensitive to overexploitation. Consequently, they have been identified as bioindicators [12,13] of management strategies implemented for fish assemblages in coastal rocky habitats [14]. Their population status is closely tied to fishing pressure, as they remain highly attractive to both professional and recreational fishers [11,15,16]. Over the long term, these species are assumed to demonstrate the "reserve effect", exhibiting increases in abundance, size, and biomass within marine protected areas (MPAs) [17–21]. They also serve as indicators of the "spillover effect", whereby individuals may move beyond reserve boundaries due to increased population density and migratory behaviors within the protected area [14,18,19,21,22]. Thus, assessing the population status of these three bioindicator species provides valuable ecosystem insights and helps guide effective conservation management strategies.

In Corsica (France), the fourth-largest island in the western Mediterranean basin *Dentex dentex*, *Epinephelus marginatus*, *Sciaena umbra* are protected through specific regulations. Since 2023, new fishing regulations for *Dentex dentex* have been introduced, including a closed season for artisanal and recreational fishers from March 15 to April 15 during the breeding period. The regulations also set a minimum catch size of 40 cm and impose a limit of one fish per recreational fisher (with a maximum of two fish per boat, [23]). *E. marginatus* and *S. umbra* are exclusively exploited by artisanal fishers, with recreational fishing being banned since 1980 and 2013 respectively [24]. The conservation efforts in Corsica are further supported by three MPAs located in the island's coastal waters [25]. The most recently implemented, the Marine Natural Park of Cape Corse and Agriate (MNP CCA) since 2016, is the focus of this study. Unlike natural reserves, which provide strong protection levels (from fully to highly protected zones), natural marine parks, as a whole, are typically classified as offering only moderate protection [26]. Within the MNP CCA, the "Nonza-San Fiurenzu" No-Take Zone (NTZ) (hereafter Nonza NTZ) is of particular importance as it is a fishing exclusion zone managed by artisanal fishers. Nonza NTZ strictly prohibits all forms of fishing and diving since 1978 to minimize any potential disturbance to fish populations. Since 2019, the MNP CCA has implemented a management plan targeting the sustainable management of *D. dentex*, *E. marginatus*, *S. umbra*, using underwater visual census (UVC) to monitor their population status.

UVC methods are non-destructive, making them particularly suitable for surveying MPAs [17]. Various techniques have been developed, such as transects, fixed points, and video

capture using scuba divers, remotely operated vehicles (ROVs), or rotating remote systems [27]. For shallow areas, however, UVC methods employing transects and fixed points are often more convenient than video methods [17,28]. UVC techniques are also relatively fast and cost effective to perform, do not require subsequent lab-work, and are useful to record a great variety of variables such as density, biomass, size structure, species composition, habitat characteristics. Despite these advantages, UVC methods have limitations, including constraints related to scuba diving (depth and time), observer experience, and species-specific reactions to divers. Consequently, it is widely accepted that UVC methods may underestimate fish abundance due to their reliance on observations within a specified area. Conversely, UVC methods can also overestimate densities at small scales by focusing on areas where fish may temporarily aggregate, without accounting for the entire habitat or the varying life stages of the species.

The primary objective of this study was to identify the NTZ protection level effectiveness within the MNP CCA by comparing the density and size structure of three bioindicator species inside and outside the park's Nonza NTZ. Also in doing so, we wanted to evaluate the effectiveness of two UVC protocols, the cross and comb protocols, by comparing their density estimates at similar sites and times. To meet these objectives, repeated diving surveys were conducted from 2018 to 2022 (488 transects). Data on density and size were analyzed via Kolmogorov-Smirnov and Generalized Linear Models to assess protection status, UVC method, temporal trends, and key environmental factors.

## Materials and methods

### Study site: The Natural Marine Park of Cap Corse and Agriate (MNP CCA)

The Natural Marine Park of Cap Corse and Agriate (MNP CCA, https://parc-marin-cap-corse-agriate.fr/), established in 2016, is the largest marine park in metropolitan France, encompassing the entire northern region of Corsica and covering an area of 6,830 km². It includes two No-Take Zones (NTZs) where all forms of fishing and scuba diving are strictly prohibited: "Capu Sagru" near Bastia (1.59 km²) and "Nonza-San Fiurenzu" (hereafter Nonza NTZ) along the west coast (24.19 km², Fig 1).

In 2018, following an initial scuba-diver assessment of potential locations, three sectors—Agriate, Canelle, and Nonza—were selected as monitoring sites within the MNP CCA (Fig 1). These areas share similar reef characteristics at depths of 15–25,m, with predominantly *Posidonia oceanica* meadows and algal-covered rocky bottoms, and less than 15% sandy or detritic substrate [29]. Additionally, the sites in Nonza (N1-N4) are located within the Nonza NTZ, while the other sectors are not subject to strict fishing restrictions.

### UVC monitoring of *Dentex dentex*, *Epinephelus marginatus*, and *Sciaena umbra*

**Cross protocol.**   From 2018 to 2022, we applied the cross UVC method at twelve rocky-reef sites distributed over three coastal sectors—Agriate (A1–A4), Canelle (C1–C4) and Nonza (N1–N4; Table 1, Fig 1). At each site at daytime and during both survey seasons (summer and autumn), two divers censused four transects arranged in a cross formation, positioned by GPS and compass following [28] (Fig 2). Each transect measured 50 m in length and 20 m in width (i.e., 1 000 m²), utilizing a measuring tape for accuracy. Before every seasonal campaign the same diver pairs performed a calibration dive with fish silhouettes of known sizes to standardise their length estimates. On survey days the research vessel approached the site at minimal speed, then shut off its engines to limit disturbance. The divers

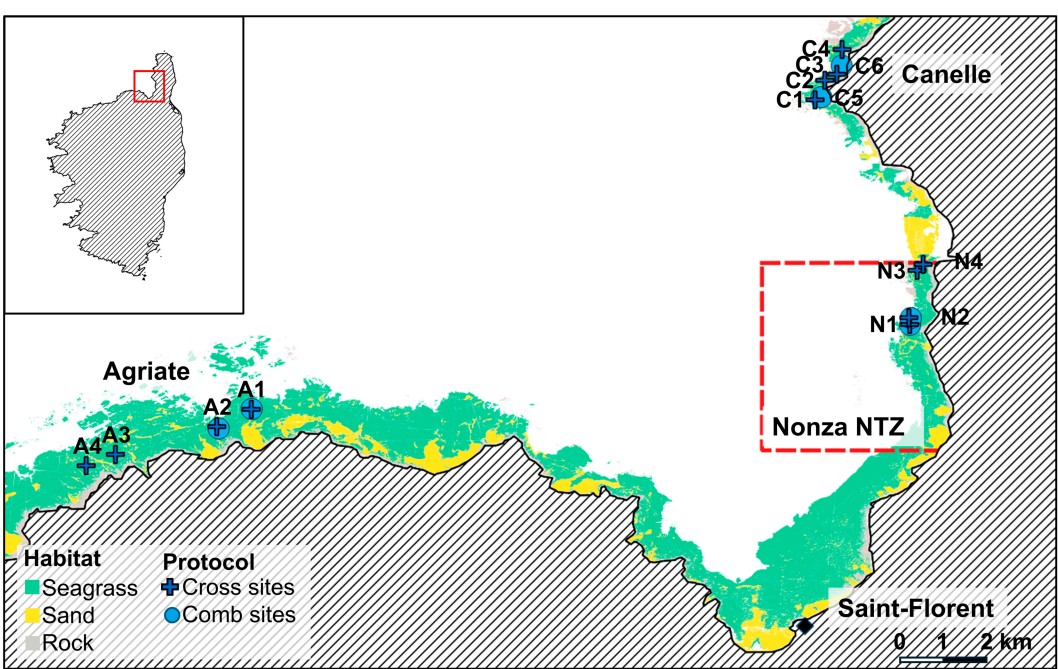

**Fig 1. Map of the northwestern coast of Corsica, showcasing the sampling sites and benthic habitats.** Sampling protocols: cross protocol sites (dark blue crosses), comb protocol sites (blue comb symbols). Monitored sectors include Agriate (A1–A4), Canelle (C1–C6), and Nonza NTZ (N1–N4, red dashed line).

**Table 1. Overview of UVC sampling performed for the Nonza, Agriate, and Canelle sites including sampling years, transect numbers, and area per transect for both comb and cross protocols.**

| Sampling Site | Cross Protocol | | | Comb Protocol | | |
|---|---|---|---|---|---|---|
| | Year | $n$ Transects | Sampling Area (m²) | Year | $n$ Transects | Sampling Area (m²) |
| **Agriate** | | | | | | |
| A1 | 18–22 | 36 | 4000 | 20–22 | 16 | 17000 |
| A2 | 18–22 | 36 | 4000 | 20–22 | 16 | 17000 |
| A3 | 18–20 | 20 | 4000 | – | – | – |
| A4 | 18–20 | 20 | 4000 | – | – | – |
| **Canelle** | | | | | | |
| C1 | 18–22 | 36 | 4000 | – | – | – |
| C2 | 18–22 | 36 | 4000 | – | – | – |
| C3 | 18–20 | 20 | 4000 | – | – | – |
| C4 | 18–20 | 20 | 4000 | – | – | – |
| C5 | – | – | – | 18–22 | 40 | 34000 |
| C6 | – | – | – | 18–22 | 40 | 34000 |
| **Nonza** | | | | | | |
| N1 | 18–22 | 36 | 4000 | 20–22 | 20 | 17000 |
| N2 | 18–22 | 36 | 4000 | 20–22 | 20 | 17000 |
| N3 | 18–20 | 20 | 4000 | – | – | – |
| N4 | 18–20 | 20 | 4000 | – | – | – |

entered the water simultaneously, conducted independent visual counts along the four transects, and recorded species identity and total length (cm) on waterproof sheets. After surfacing, they compared and merged their observations to reduce inter-observer bias. A total of

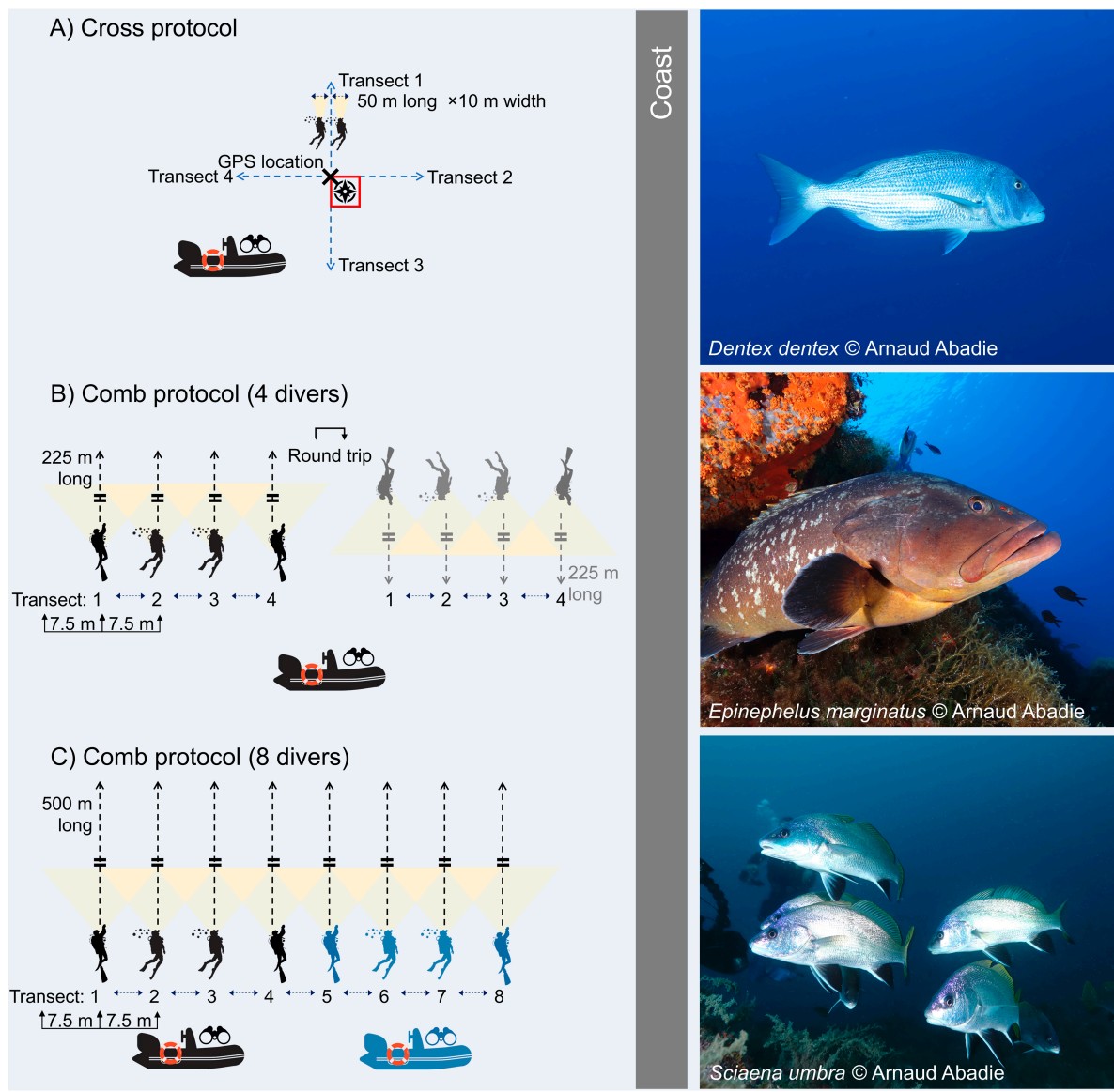

**Fig 2. Survey protocols and target species. Left panels:** layouts of the underwater visual-census transects: (A) cross protocol with two divers ($4 \times 50$ m $\times 10$ m strips); (B) comb protocol with four divers (225 m corridor, round-trip); (C) comb protocol with eight divers (single 500 m corridor). **Right panels:** photographs of the three focal predators recorded during the survey—common dentex *Dentex dentex* (top), dusky grouper *Epinephelus marginatus* (middle) and brown meagre *Sciaena umbra* (bottom).

336 transects were completed over the study period (Table 1). Budgetary and logistical constraints led us to halve the spatial coverage in 2021 and 2022; hence sites A3, A4, C3, C4, N3 and N4 were temporarily excluded.

**Comb protocol.**  From 2020 to 2022, we applied the "comb" UVC method—an adaptation of the GEM inventories [30]—at six rocky-reef sites spread over three coastal sectors: Agriate (A1, A2), Canelle (C5, C6) and Nonza (N1, N2; Table 1, Fig 1). At each site at daytime and during both survey seasons (summer and autumn), teams of either four or eight divers advanced in parallel lines, forming a comb-like corridor (Fig 2). Divers maintained a

spacing of 5–10 m depending on visibility, with 7.5 m taken as the standard width for density calculations.

- **Four-diver configuration (Agriate and Nonza).** The team followed a 250 m linear transect, then returned along a new, parallel line (i.e., round trip). Divers were dropped precisely on the start GPS coordinate, and a single vessel shadowed them at low speed to limit disturbance.
- **Eight-diver configuration (Canelle).** To cover the more heterogeneous seabed, eight divers surveyed a one-way 500 m corridor in a single pass. Two vessels accompanied the formation, each proceeding slowly behind one half of the line.

Each diver received an individual compass heading, while the most experienced pair (first and last in the line) maintained overall orientation; buddy pairs regrouped first if the formation became fragmented. As with the cross protocol, divers recorded species identity and total length (cm) on waterproof sheets, after completing a pre-season calibration dive with fish silhouettes of known sizes. The same diver teams were kept as constant as possible across years to minimise inter-observer bias, and observations were compared immediately after each dive to avoid double-counting. In total, 152 comb transects were completed during the study period (Table 1). Adverse sea conditions prevented the Agriate sector from being sampled in summer 2022.

## Data analysis

**GLM framework for density and biomass analysis.** We quantified fish populations using data from underwater visual censuses (UVCs), calculating both population density (individuals per 1000 m$^2$) and biomass density (g per 1000 m$^2$) for each of the three focal species. Biomass estimates were derived from the observed density data by incorporating species-specific size information, providing a more integrative measure that reflects both abundance and individual size structure. For each species, biomass was estimated using the standard length–weight relationship:

$$W = a \times L^b,$$

where $W$ is the estimated weight (in grams), $L$ is the body length (approximated by the midpoints of observed length classes), and the parameters $a$ and $b$ are species-specific coefficients sourced from FishBase (*D. dentex*: $a = 0.01122$, $b = 3.03$; *S. umbra*: $a = 0.00851$, $b = 3.08$; *E. marginatus*: $a = 0.01047$, $b = 3.06$). For each transect, we calculated total biomass as the sum across all length classes of the product between class-specific abundance and estimated weight. Biomass values were scaled by the survey area to obtain biomass densities.

To account for heteroscedasticity and ensure approximate normality, both population and biomass densities were transformed using $\log(x + 1)$ [33]. We then fitted Gaussian generalized linear models (GLMs) with identity link functions to assess the influence of key environmental and management factors. Fixed effects in the models included protection status (protected vs. unprotected), year, season (autumn vs. summer), sea surface temperature (SST in °C), and Chlorophyll-*a* concentration (mg/m$^3$), as well as the UVC protocol (cross vs. comb) to account for potential observer or methodological differences. We retained both SST and Chlorophyll-*a* in the models after confirming low collinearity (Pearson's $r < 0.7$). Environmental data were sourced from NASA's Earth Observations platform (https://neo.gsfc.nasa.gov/), with SST obtained from the MODIS-Aqua instrument and Chlorophyll-*a* derived using

the three-band algorithm for oligotrophic waters [31]. Both datasets had a spatial resolution of ~1 km and were averaged over 8-day intervals, then extracted and processed in QGIS (version 3.28.7) for spatial alignment with transect locations. All models were implemented in R using the base `glm()` function with a Gaussian family specification.

In preliminary analyses, we tested for potential habitat-driven differences among the three sectors (Agriate, Canelle, Nonza) by including "sector" as a factor in Kruskal-Wallis tests and PERMANOVA. These analyses revealed no significant sector-level effect on fish densities beyond what was already explained by the "protected vs. unprotected" classification. Consequently, for clarity, we focus on the NTZ (Nonza) versus the fished sectors (Agriate and Canelle), confirming that any observed reserve effect is not confounded by underlying habitat variations.

**Size distribution analysis.** Additionally, we explored significant differences in the size distribution of the three species across sectors with different protection status. Comparisons were made between the Nonza sector, which has a NTZ status, and the other sectors where fishing is allowed. Measurements were categorized into fourteen ranges: 1) 10–15 cm; 2) 16–20 cm; 3) 21–25 cm; 4) 26–30 cm; 5) 31–35 cm; 6) 36–40 cm; 7) 41–45 cm; 8) 46–50 cm; 9) 51–55 cm; 10) 56–60 cm; 11) 61–65 cm; 12) 66–70 cm; 13) 71–75 cm; and 14) > 76 cm. For this analysis, we performed non-parametric two-sample Kolmogorov-Smirnov (K-S) tests [33] using the "`ks.test()`" function implemented in the R "stats" package [32]. The K-S test compares the cumulative distributions of two datasets to determine if they differ significantly. The size distribution of *D. dentex*, *E. marginatus*, and *S. umbra* across different protection status (protected vs unprotected) were further illustrated in bar plots showing the proportion of individuals in each size class, thereby highlighting differences in size frequency distributions between protected and unprotected areas for each species.

## Results

From 2018 to 2022, underwater visual censuses (UVCs) recorded a total of 1 194 individuals of *Dentex dentex* with an average density of $1.39 \pm 0.21$ individuals $1000 \cdot m^{-2}$ a corresponding biomass of $632 \pm 101$ g$\cdot 1000 \cdot m^{-2}$. The most represented size class for *D. dentex* was 31–35 cm. In the same period, 350 individuals of *Epinephelus marginatus* were observed, giving an average density of $0.46 \pm 0.05$ individuals $1000 \cdot m^{-2}$ and a biomass of $885 \pm 105$ g$\cdot 1000 \cdot m^{-2}$; the modal size class was 41–45 cm. Finally, 394 individuals of *Sciaena umbra* were counted, corresponding to an average density of $0.43 \pm 0.13$ ind $\cdot 1000$ m$^{-2}$ and a biomass of $151 \pm 36$ g $\cdot 1000$ m$^{-2}$. The most frequent size class for this species was 26–30 cm.

### Effect of marine protection on fish density and biomass

Protection status did not emerge as a significant predictor of *D. dentex* density: mean values were comparable between the unprotected sites ($1.53 \pm 0.29$ ind$\cdot 1000$ m$^{-2}$) and the Nonza NTZ ($1.10 \pm 0.14$ ind$\cdot 1000$ m$^{-2}$; Table 2, Figs 3 and 4). By contrast, biomass displayed a significant reserve effect, with higher values inside the NTZ ($678.88 \pm 140.91$ g$\cdot 1000$ m$^{-2}$) than outside ($632.15 \pm 101.26$ g$\cdot 1000$ m$^{-2}$) (Table 3).

For *E. marginatus*, protection status was the *strongest* GLM predictor (Table 2, Figs 3 and 4). Densities reached $1.01 \pm 0.12$ ind$\cdot 1000$ m$^{-2}$ in the Nonza NTZ versus $0.22 \pm 0.03$ ind$\cdot 1000$ m$^{-2}$ outside. The pattern was even more pronounced for biomass: $2\,061.41 \pm 279.72$ g$\cdot 1000$ m$^{-2}$ in the NTZ compared with only $352.47 \pm 67.70$ g$\cdot 1000$ m$^{-2}$ in fished areas (Table 3).

Protection status likewise dominated the models for *S. umbra*, yielding higher densities in the NTZ ($0.94 \pm 0.38$ ind$\cdot 1000$ m$^{-2}$) than in unprotected sectors ($0.20 \pm 0.06$ ind$\cdot 1000$ m$^{-2}$) (Table 2, Figs 3 and 4). For biomass, protection was identified as the second most influential

**Table 2. GLM results for population density of *Dentex dentex*, *Epinephelus marginatus*, and *Sciaena umbra*, showing the effects of protection status, year, season, sea surface temperature (SST), and Chlorophyll-*a* concentration. Significance is indicated by \*, \*\*, and \*\*\*, representing $p \leq 0.05$, $p \leq 0.01$, and $p \leq 0.001$, respectively.**

| Variable | *D. dentex* | | | | *E. marginatus* | | | | *S. umbra* | | | |
|---|---|---|---|---|---|---|---|---|---|---|---|---|
| | Slope | Std. Error | *p*-Value | Sig. | Slope | Std. Error | *p*-Value | Sig. | Slope | Std. Error | *p*-Value | Sig. |
| Protocol: Cross-Comb | −0.023 | 0.082 | 0.775 | | −0.154 | 0.051 | 0.003 | \*\* | 0.108 | 0.053 | 0.041 | \* |
| Site: Protected-Unprotected | −0.055 | 0.065 | 0.401 | | −0.364 | 0.041 | 7.270e−18 | \*\*\* | −0.172 | 0.042 | 5.115e−05 | \*\*\* |
| Year | −0.006 | 0.029 | 0.848 | | 0.013 | 0.018 | 0.482 | | 0.008 | 0.019 | 0.669 | |
| Season: Autumn-Summer | −0.305 | 0.085 | 3.615e−04 | \*\*\* | −0.090 | 0.053 | 0.089 | | 0.090 | 0.055 | 0.099 | |
| SST [°C] | −0.054 | 0.018 | 0.004 | \*\* | 0.016 | 0.012 | 0.169 | | −0.035 | 0.012 | 0.003 | \*\* |
| Chlorophyll-*a* [mg/m$^3$] | −2.924 | 1.905 | 0.125 | | −1.358 | 1.187 | 0.253 | | −1.678 | 1.226 | 0.172 | |

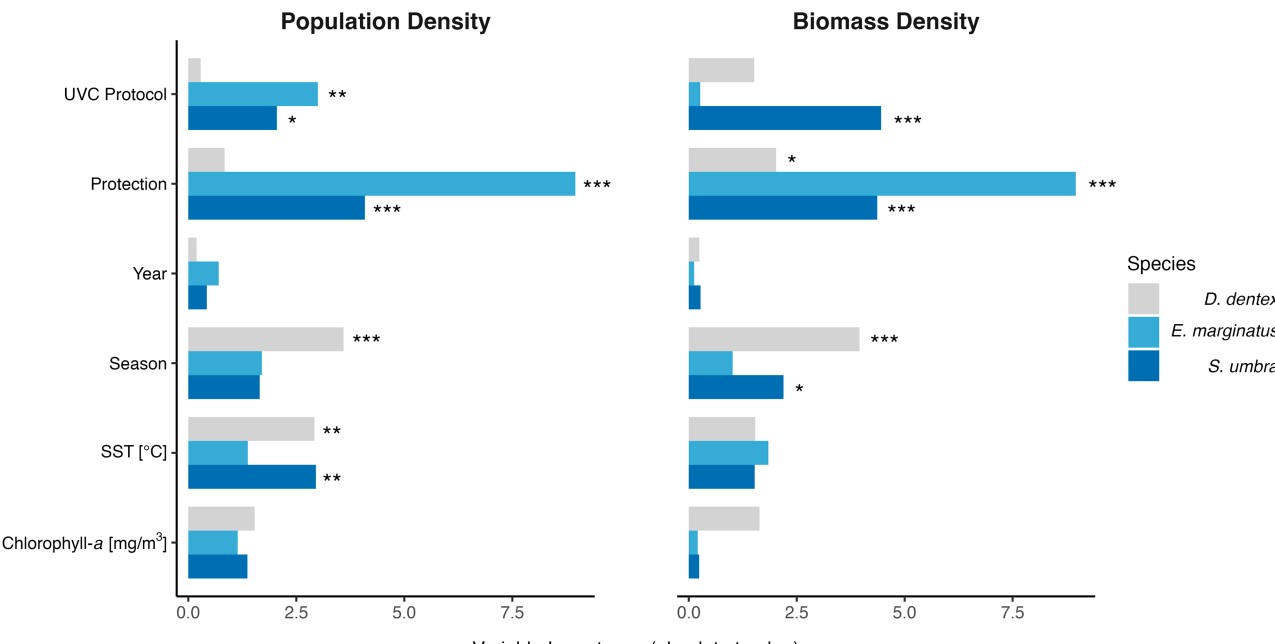

**Fig 3. Variable importance of predictors in Gaussian GLMs for *Dentex dentex*, *Epinephelus marginatus*, and *Sciaena umbra*, based on the absolute value of the t-statistics.** Panels show results for models fitted to log-transformed population density (left) and biomass density (right). Predictors include UVC Protocol (cross vs. comb), Protection status (NTZ vs. unprotected), Year (2018–2022), Season (autumn vs. summer), Sea Surface Temperature (SST, °C), and Chlorophyll-*a* concentration (mg/m$^{-3}$). Significance levels in the main text are indicated as \*, \*\*, and \*\*\*, representing $p \leq 0.05$, $p \leq 0.01$, and $p \leq 0.001$, respectively.

factor: $324.95 \pm 94.59$ g·1000 m$^{-2}$ inside the NTZ versus $72.43 \pm 28.47$ g·1000 m$^{-2}$ outside (Table 3).

## Yearly and seasonal changes in density

The density of *D. dentex* showed no significant differences between years, ranging from $0.83 \pm 0.14$ ind·1000 m$^{-2}$ in 2022 to $4.15 \pm 1.60$ ind·1000 m$^{-2}$ in 2018 (Table 2, Fig 5). Consistently, biomass did not vary significantly among years, oscillating between $339.96 \pm 83.26$ g·1000 m$^{-2}$ in 2020 and $1\,898.15 \pm 736.67$ g·1000 m$^{-2}$ in 2018. However, season emerged as the most significant variable (Table 2, Fig 3), with summer surveys detecting lower densities ($0.54 \pm 0.09$ ind·1000 m$^{-2}$) than the corresponding ones conducted

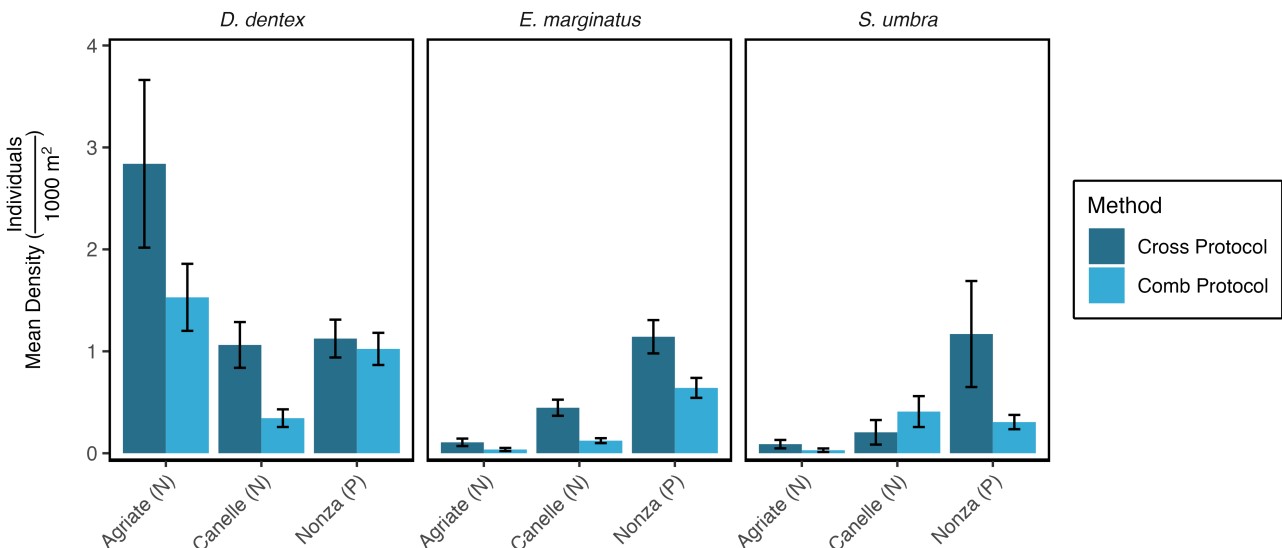

**Fig 4. Population density of the fish species across the three sites and two UVC protocols.** Agriate and Canelle are unprotected (N) while Nonza is protected (P), designated as a No-Take Zone (NTZ). Error bars represent the standard error of the mean.

**Table 3. GLM results for biomass density of *Dentex dentex*, *Epinephelus marginatus*, and *Sciaena umbra*, showing the effects of protection status, year, season, sea surface temperature (SST), and Chlorophyll-*a* concentration. Significance is indicated by \*, \*\*, and \*\*\*, representing $p \leq 0.05$, $p \leq 0.01$, and $p \leq 0.001$, respectively.**

| Biomass density | D. dentex | | | | E. marginatus | | | | S. umbra | | | |
|---|---|---|---|---|---|---|---|---|---|---|---|---|
| | Slope | Std. Error | p-Value | Sig. | Slope | Std. Error | p-Value | Sig. | Slope | Std. Error | p-Value | Sig. |
| Protocol: Cross-Comb | 0.584 | 0.386 | 0.130 | | 0.104 | 0.394 | 0.792 | | 1.174 | 0.264 | 1.047e−05 | *** |
| Site: Protected-Unprotected | −0.618 | 0.306 | 0.044 | * | −2.805 | 0.313 | 6.968e−18 | *** | −0.913 | 0.209 | 1.560e−05 | *** |
| Year | 0.034 | 0.138 | 0.808 | | −0.018 | 0.141 | 0.901 | | −0.026 | 0.094 | 0.786 | |
| Season: Autumn-Summer | −1.575 | 0.398 | 8.894e−05 | *** | −0.414 | 0.408 | 0.310 | | 0.597 | 0.272 | 0.029 | * |
| SST [°C] | −0.133 | 0.087 | 0.125 | | 0.163 | 0.089 | 0.066 | | −0.090 | 0.059 | 0.128 | |
| Chlorophyll-*a* [mg/m$^3$] | −14.627 | 8.936 | 0.102 | | −1.888 | 9.141 | 0.836 | | −1.463 | 6.109 | 0.811 | |

in autumn ($1.99 \pm 0.34$ ind·1000 m$^{-2}$). Biomass mirrored this seasonal pattern, reaching $335.39 \pm 74.74$ g·1000 m$^{-2}$ in summer and $838.22 \pm 162.57$ g·1000 m$^{-2}$ in autumn (Table 3).

Similarly, no temporal trend was observed for the density of *E. marginatus* across the years, ranging from $0.40 \pm 0.08$ ind·1000 m$^{-2}$ in 2022 to $0.52 \pm 0.12$ ind·1000 m$^{-2}$ in 2019 (Table 2, Fig 5). Biomass likewise showed no significant inter-annual variation, fluctuating between $776.84 \pm 225.56$ g·1000 m$^{-2}$ in 2022 and 1 082.67 $\pm$ 276.52 g·1000 m$^{-2}$ in 2019. Densities were statistically similar between autumn ($0.47 \pm 0.06$ ind·1000 m$^{-2}$) and summer ($0.46 \pm 0.07$ ind·1000 m$^{-2}$), and biomass did not differ significantly between seasons either ($940.46 \pm 175.73$ g·1000 m$^{-2}$ in summer versus $846.09 \pm 129.55$ g·1000 m$^{-2}$ in autumn) (Table 3).

For *S. umbra*, the GLM highlighted no significant temporal fluctuations in density (Table 2). Neither the year, increasing from $0.17 \pm 0.09$ ind·1000 m$^{-2}$ in 2018 to a peak of $1.02 \pm 0.46$ ind·1000 m$^{-2}$ in 2020, nor the season (autumn: $0.49 \pm 0.21$ ind·1000 m$^{-2}$; summer: $0.34 \pm 0.08$ ind·1000 m$^{-2}$) were significant predictors of density (Fig 5). Biomass likewise exhibited no inter-annual variation, ranging from $66.90 \pm 34.99$ g·1000 m$^{-2}$ in 2018 to $306.99 \pm 122.92$ g·1000 m$^{-2}$ in 2019, but showed a slight though significant seasonal effect,

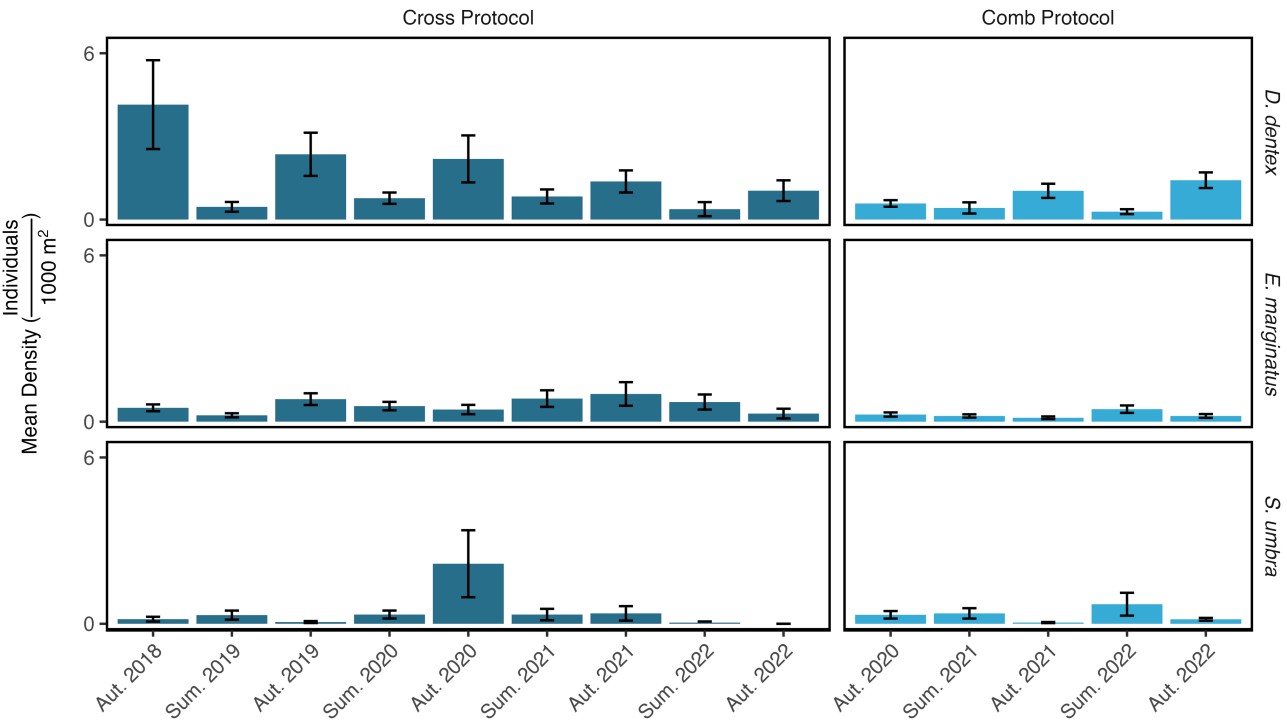

**Fig 5. Mean density (individuals per 1000 m²) of *Dentex dentex*, *Epinephelus marginatus*, and *Sciaena umbra* across seasons and years, using both comb and cross UVC protocols.** Error bars represent the standard error of the mean.

being lower in summer (145.72 ± 33.59 g·1000 m⁻²) than in autumn (154.81 ± 55.91 g·1000 m⁻²) (Table 3).

## Influence of environmental covariates

Among the environmental predictors tested, sea-surface temperature (SST) was the only variable that entered the density models as significant, and this occurred solely for *D. dentex* and *S. umbra*: in both cases densities declined as SST increased (Table 2, Fig 3). SST was not retained for *E. marginatus* and showed no detectable influence on biomass for any of the three species. Over the 2018–2022 period SST exhibited a pronounced seasonal signal, spanning 17.05–25.39 °C in autumn and 22.18–26.55 °C in summer. Chlorophyll-*a* concentration, by contrast, had no significant effect on either density or biomass for *D. dentex*, *E. marginatus*, or *S. umbra*.

## Variations in fish size distribution

The Kolmogorov-Smirnov tests (KS) indicated no statistically significant difference in the observed size distribution of the three fish species between protected and unprotected localities (Fig 6). For *D. dentex*, 48.3% of individuals in protected areas were at or above the size of maturity (31 to 35 cm [11]), which was very similar to the proportion in unprotected areas (48.7%). Similarly, for *E. marginatus*, 54.4% of individuals in protected areas had this size of maturity or larger (46 to 50 cm) [34], while in unprotected areas this was only 36.7%. For *S. umbra*, 81.8% of individuals in protected areas were at or above the size of maturity (21 to 25 cm [35]), in contrast to 48.9% in unprotected areas.

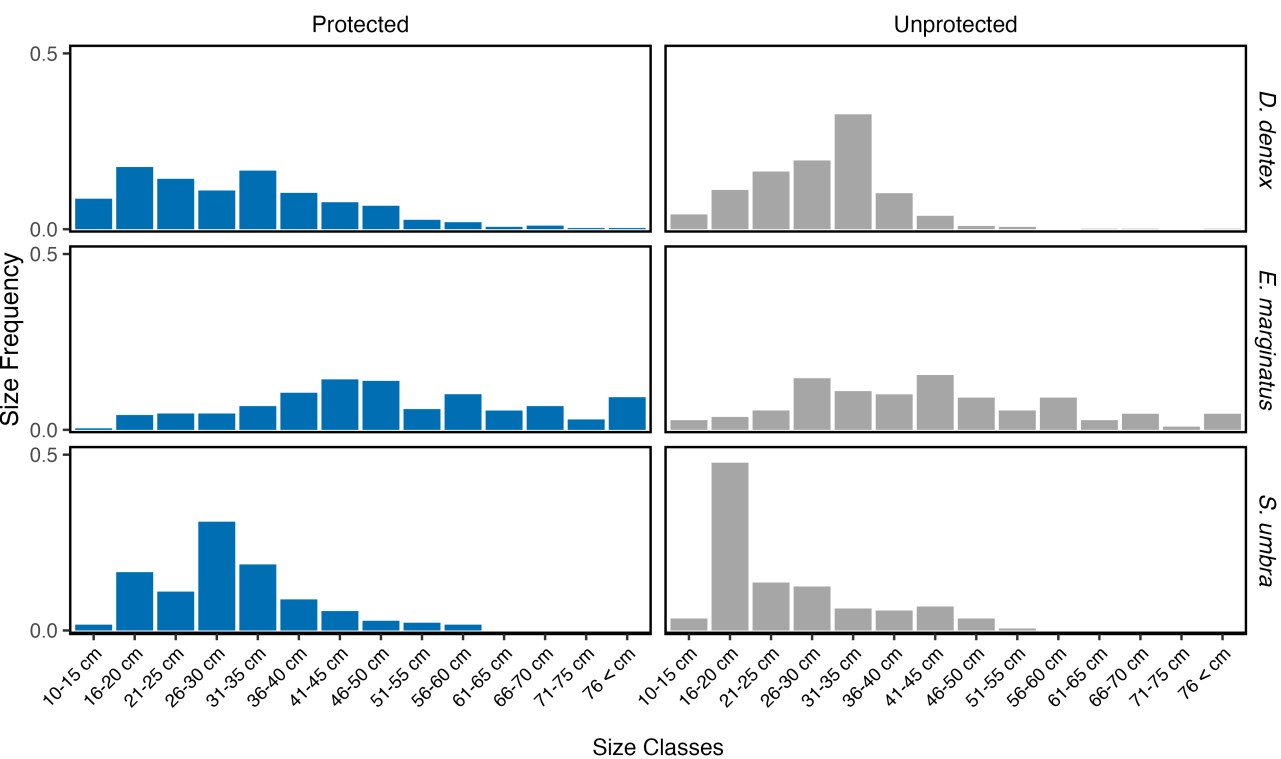

**Fig 6. Size frequency distributions of *D. dentex*, *E. marginatus*, and *S. umbra* across different protection statuses (Protected vs Unprotected).** Each bar represents the proportion of individuals within a specific size class. *P-values* of Kolmogorov-Smirnov tests investigating the effect of protection level on size distribution are overlaid for each species.

### Effect of UVC protocol on density and biomass estimates

For *D. dentex*, protocol choice had no detectable influence on either metric: mean density varied from 0.77 to 1.68 ind · 1000 m$^{-2}$, and biomass from 321 ± 102 to 773 ± 101 g · 1000 m$^{-2}$ (Table 2, Fig 4). In contrast, protocol was the second-strongest predictor of *E. marginatus* density. Counts obtained with the cross protocol (0.57 ± 0.07 ind·1000 m$^{-2}$) were more than double those from the comb protocol (0.24 ± 0.03 ind·1000 m$^{-2}$; Figs 3 and 4). Biomass, however, did not differ significantly between protocols (1 069 ± 145 vs. 477 ± 96 g · 1000 m$^{-2}$). For *S. umbra* the protocol effect was even clearer: densities averaged 0.49 ± 0.18 ind·1000 m$^{-2}$ with the cross survey and 0.30 ± 0.08 ind · 1000 m$^{-2}$ with the comb survey, while biomass reached 180 ± 51 and 86 ± 20 g · 1000 m$^{-2}$, respectively.

## Discussion

### Effectiveness of no-take zones in enhancing fish populations

The similar densities of *Dentex dentex* inside and outside the reserve contrast with a significant biomass increase in the Nonza NTZ, showing that protection favours larger individuals rather than more individuals. The lack of a density signal likely stems from the species' mobility: as a bentho-pelagic predator it can move well beyond the 5-km$^2$ NTZ [11,36,37]. Smaller, more enclosed MPAs have detected clear density differences—e.g. 2.0 to 0.2 ind·1000 m$^{-2}$ across zones in the 1.3-km$^2$ Medes reserve [38] and higher CPUE near the Bonifacio Strait NTZ [39]—implying that reserve size and landscape context modulate

the response. In our survey, densities peaked in Agriate, where extensive *Posidonia oceanica* meadows offer preferred nursery habitat for juveniles [15]. We therefore suggest that habitat availability drives juvenile abundance, whereas protection chiefly benefits adult size structure—hence biomass—inside the NTZ. Because wide-ranging adults frequently exit small reserves [40,41], NTZs should be embedded in an ecologically coherent MPA network that represents both nursery and adult habitats [42].

In contrast, protection status was the strongest predictor for *E. marginatus*: mean density in the Nonza NTZ reached 1.01 ind·1000 m$^{-2}$, at least three times higher than in the fished Agriate and Canelle sectors (<0.31 ind·1000 m$^{-2}$). This pronounced reserve effect reflects the species' life history—high longevity, slow growth and monandric protogynous hermaphroditism—and its sedentary, territorial behaviour with strong site fidelity and spawning aggregations, traits that heighten susceptibility to fishing [34,43,44]. NTZs give individuals time to attain large, reproductively active sizes, thereby increasing local abundance and demographic stability [40,45,46]. Comparable responses have been documented across the north-western Mediterranean. Integral reserves at Cabo de Palos, the Medes Islands, Portofino, Tavolara–Punta Coda Cavallo, Scandola and the Cabrera Archipelago typically harbour the highest densities of *E. marginatus* (e.g. 8–24 ind·1000 m$^{-2}$ in Cabrera versus 0.4–2.0 outside; [13,18,28,47,48]). The range observed at Nonza is consistent with estimates for Tavolara (Sardinia, Italy; 4–6.2 ind·1000 m$^{-2}$) and the Bonifacio Strait Reserve (Corsica, France; 1.5–2.5 ind·1000 m$^{-2}$) [48,49], confirming that well-enforced NTZs in Corsica provide benefits on par with older Mediterranean MPAs.

Protection status was a strong predictor of *S. umbra* density: counts in the Nonza NTZ averaged 0.94 ind·1000 m$^{-2}$, more than triple those in the unprotected Agriate and Canelle sectors (both <0.29 ind·1000 m$^{-2}$). This sharp contrast reflects the species' small home range (<1 km$^2$) and limited dispersal, which heighten vulnerability to local fishing pressure [50–52]. Strong site fidelity during reproduction further magnifies the benefit of spatial protection, allowing populations to rebound rapidly in suitable habitat [40,41]. Our findings agree with previous Mediterranean surveys. In fully protected zones, [53] recorded *S. umbra* vocal activity at 70% of sites versus 30% in partially protected areas and 45% outside MPAs, while densities in the Scandola Reserve reached 3.42 ± 9.81 ind·1000 m$^{-2}$ in the NTZ and only 0.20 ± 1.68 outside [54]. Similar reserve effects have been reported for the Gulf of Lion, the Calanques National Park, and Tavolara–Punta Coda Cavallo [53,54].

Our results demonstrate that a well-enforced no-take zone (NTZ) embedded in a multiuse MPA can generate a reserve effect comparable to that documented in fully protected areas [13,14]. Biomass gains for the three focal predators have direct functional consequences: the keystone *Dentex dentex* and the apex predator *Epinephelus marginatus* exert top-down control on pelagic fishes such as *Seriola dumerili*, while the lower-trophic *Sciaena umbra* is itself regulated by piscivores [11,34,55,56]. Therefore, the protection of these species within MPAs not only benefits their populations but also contributes to the overall resilience and stability of the marine ecosystems they inhabit [13,14,57].

## Temporal fluctuations in density estimates

In terms of inter-annual variability, our results covering the period from 2018 to 2022, showed no significant differences in the density and biomass estimates of *D. dentex*, *E. marginatus*, and *S. umbra* between the years, suggesting stability in the populations observed in the MNP CCA. In Corsica, these findings align with fisheries data from [16], who found no significant population trend for *D. dentex* catch per unit effort (CPUE) from 2011 to 2020. However, [58] reported a global decrease in the proportion of carnivorous species and

large individuals, including our three targeted species, in the Bay of Calvi, Corsica, between 2012 and 2017. These findings suggest that while some species may not show significant inter-annual density fluctuations, broader changes in community structure and composition are evident. These changes are likely driven by environmental and anthropogenic factors [2], emphasizing the importance of continued monitoring to understand and mitigate impacts on coastal fish communities.

Delving into seasonal variations, the observed *D. dentex* density and biomass estimates varied with the sampling season, being almost significantly higher in autumn than in summer. The summer period is probably unfavorable for observing this species as it corresponds to the breeding period, during which *D. dentex* potentially moves further offshore into deeper areas [11]. This observation aligns with [59], who found that juveniles (17–25 cm TL) first appear in autumn–winter, whereas spring surveys often record large adults engaged in pre-spawning aggregations. In summer, medium-sized individuals (1–5 kg) are often observed around rocky outcrops at depths between 10 and 35 meters [11,15]. Overall, our results reinforce the need to sample both seasons to capture the full demographic signal of *D. dentex*.

For *E. marginatus*, our GLMs detected no significant seasonal shift in either density or biomass, in contrast with the pronounced summer peaks reported for other Mediterranean MPAs. In the Bonifacio Strait Natural Reserve (BSNR, Corsica) densities rose from 1.5 to 2.5 ind·1000 m$^{-2}$ between autumn and summer [49], while at the Medes Islands Reserve (Spain) they climbed from ∼5 ind·1000 m$^{-2}$ in winter to ∼40 ind·1000 m$^{-2}$ at the height of the spawning season, then fell back to ∼15 ind·1000 m$^{-2}$ in autumn [60]. Such pulses coincide with the reproductive period (May–September), when dominant males defend shallow ledges and pairs perform brief upward sorties at dusk to spawn in aggregations [34,44,48]. Two factors may explain why this pattern was absent from our survey: (i) our transects did not intersect the shallow breeding ledges (<25 m) where spawning aggregations form, and (ii) post-spawning dispersal to neighbouring reefs likely smooths out seasonal contrasts at the broader spatial scale we sampled.

For *S. umbra* we found no seasonal change in density, whereas biomass was higher in autumn. During summer the species disperses slightly to seek mates, yet its home range remains small (<1 km$^2$), limiting detectable shifts in abundance [51,52]. Earlier work in Port-Cros (France) reported modest density differences between spring (0.16–0.36 ind·1000 m$^{-2}$) and autumn (0.13–0.25 ind·1000 m$^{-2}$) [61], and a 47% rise in total counts from spring to autumn by 2005 [62]. These mixed results highlight how little is known about *S. umbra* spawning sites and movements in Corsica; targeted tracking during the breeding season is needed to clarify any seasonal redistributions [15].

## Environmental drivers of fish population density

We found that temperature (SST) has a significantly negative impact on the density of *Dentex dentex* and *Sciaena umbra* aligning with [15], who emphasized that temperature is a predictive variables for the habitats of those two species. Temperature is a key environmental factor that profoundly influences the fitness and survival of ectothermic organisms through physiological effects [63,64]. It affects not only metabolic rates but also reproductive behavior, feeding activity, and even calling activity [36,53]. Ectothermic organisms typically thrive within a limited temperature range that maximizes their physiological performance [63,64]. Deviations from this optimal range can lead to reduced growth, reproduction, foraging efficiency, and competitiveness. The negative relationship observed for *D. dentex* and *S. umbra* may be explained by the fact that these species have a cooler thermal optimum,

particularly in a system where temperatures can reach high levels (up to 26.6°C during the UVC surveys, [65–67]). Shifts away from this thermal optimum can drive species to seek thermal refuges in deeper habitats according to changes in the thermocline, making them less detectable in shallower UVC survey sites [36,68,69]. This vertical migration of *D. dentex* and *S. umbra* might also be influenced indirectly by the distribution of their preys, suggesting the need for monitoring adjacent deeper sites to better understand these dynamics. Additionally, we assume that *E. marginatus* showed no temperature-dependent distribution patterns because it is a less mobile and more territorial species [44]. Indeed, small-scale migrations in habitat depth are only visible on a seasonal scale, when the seawater temperature exceeded 20 °C, during which *E. marginatus* breed [44].

Our GLMs detected no significant effect of chlorophyll-*a*—used here as a proxy for primary production—on any of the response variables. This contrasts with the basin-scale habitat model of [15], in which chlorophyll-*a* strongly influenced the distributions of *D. dentex* and *S. umbra*. The eastern Mediterranean Sea is characteristically oligotrophic [70]: nutrient scarcity constrains primary production, and bottom-up effects can cascade to top predators. It is therefore plausible that the coarser spatial and temporal resolution of our data reduced the statistical power needed to detect such relationships.

## Size distribution and reserve effect

One of the main expectations for NTZs is a size-selective retention of the most competitive individuals: by removing fishing pressure, NTZs allow individuals to survive longer and grow larger. Inside an NTZ, resources remain finite, so intense intraspecific competition can prompt smaller fish to spill over to adjacent fished grounds, whereas the largest, most competitive—and therefore most fecund—individuals tend to stay and capitalise on local resources [21]. The magnitude of this reserve effect depends on MPA attributes such as age, size, shape, habitat heterogeneity and enforcement level [40,71,72].

Our biomass-based GLMs reveal a significant reserve effect for all three species, supporting higher biomasses in NTz (g·1000 m$^{-2}$) than adjacent fished areas. This pattern implies that individuals are, on average, larger inside the NTZ, a result in line with the higher proportions of mature-sized fish that we observed for *E. marginatus* and *S. umbra*. The apparent mismatch between significant biomass differences and non-significant K-S tests may be attribute to weight scales allometrically with length; a modest enrichment in larger fish therefore boosts biomass disproportionately while contributing little to the rank-based KS statistic. Besides, the K-S test compares the whole distribution, whereas the biomass signal is driven mainly by the upper tail. Thus, a subtle enrichment in the largest individuals raises biomass but contributes little to the rank-based statistic.

For *D. dentex*, the scarcity of small individuals likely reflects our focus on depths >15 m, whereas juveniles (20–50 mm TL, June–August) occupy shallow nursery zones (<5 m depth) at the edge of *Posidonia oceanica* meadows or within crevices and caves [11]. Juveniles of *E. marginatus* are similarly elusive [73,74], and their nurseries remain poorly documented. The cryptic, mainly nocturnal behaviour of juvenile *S. umbra*, along with their migration to very shallow coastal environments [75,76] makes their daytime observation quite rare. Consequently, our survey design was not optimised to census the youngest size classes, a limitation that probably reduced the sensitivity of the KS tests.

Earlier studies have consistently reported a reserve effect for *E. marginatus*—with mean size declining from NTZs to fished areas and intermediate sizes in buffer zones [18,21]—and our biomass findings corroborate this trend. We therefore interpret the significant biomass enrichment in the NTZ as evidence that protection is already benefiting individual

growth for all three species, even if size-class K-S statistics do not yet fully reflect this difference. To strengthen future assessments we recommend supplementing standard UVCs with protocols tailored to shallow-water nurseries and cryptic juveniles, thereby capturing the complete life-history spectrum of *D. dentex*, *E. marginatus* and *S. umbra*.

## Variability in UVC protocols and its impact on density estimates

Our analysis revealed that the choice of UVC protocol significantly affected density estimates for *S. umbra* and both density and biomass estimates for *E. marginatus*. Specifically, both species densities were notably higher when using the cross protocol compared to the comb protocol. This discrepancy can be attributed to the habitat preferences and behaviors of *E. marginatus* and *S. umbra*. As a reef-associated species, these species often inhabits complex environments with crevices and caves [34]. Conversely, as noted by [77], larger transects can reduce bias by capturing a broader ecological context and compensating for low densities (i.e., the dilution effect). The comb protocol, which involves more divers (4 to 8) spaced further apart, may frighten the species, causing them to hide and potentially leading to lower detection rates, resulting in underestimated density estimates [80]. The cross protocol, involving only two divers in a more focused area with reduced disturbance, may be particularly effective in detecting hidden or sedentary species. However, its spatial design could lead to some overestimation, especially for slow-moving species like *E. marginatus*, as individuals may be reobserved within the core area, resulting in potential double-counting. *S. umbra*, largely nocturnal [54,69], remains difficult to census by day under either protocol, but the cross protocol still supported higher detections. For *D. dentex*—a mobile, demersal predator that forages in more open substrate—both protocols returned comparable densities, suggesting that observer configuration is less critical for wide-ranging species [11,36].

The observed differences align with recommendations by [28], who advocate for combining large and medium transects to account for different size classes and behaviors. This approach enhances the accuracy of density estimates across species with varying habitat preferences and behavioral traits. Therefore, to improve the reliability of UVC surveys, we suggest reducing the simultaneous counting of species with different behaviors and habitat preferences in the water column [78,79]. Instead, grouping species with similar behaviors and tailoring survey protocols to these groups can lead to more accurate population assessments. For example, [78] found that UVC recorded more individuals and higher species richness compared to baited remote underwater video (BRUV), particularly for herbivores and territorial species, while BRUV was more effective for mobile predators. Similarly, [79] observed that UVC provided better estimates of rare or cryptic species, whereas BRUV offered increased coverage for diver-averse species and insights into species behavior. Additionally, one potential improvement for the detection of *S. umbra* would be to conduct surveys earlier in the day, at or near sunrise as it could provide better opportunities to detect these elusive species.

Finally, the comb protocol, requiring up to eight divers plus an additional boat and skipper, significantly increases manpower, fuel consumption, and scheduling complexity. In contrast, the two-diver cross protocol (with a single skipper) is generally more cost-effective and simpler to organize, though in theory it could miss certain individuals. Nevertheless, our results indicate that the cross protocol detected similar or even higher numbers of *E. marginatus* and *S. umbra* than the comb protocol. Consequently, cost-effectiveness and diver availability become crucial factors when selecting a UVC method, particularly for long-term monitoring programs.

## Conclusion

**Role of MPAs in species protection and monitoring**. Our findings showed how a newly moderate-protection parks, when coupled with carefully managed NTZs, can effectively bolster vulnerable fish populations. These protected areas provide refuge from fishing pressures, allowing to grow larger and reach reproductive maturity, as observed for *D. dentex*, *E. marginatus*, and *S. umbra*. However, the *D. dentex* densities not responding to protection status indicate a limited effectiveness of smaller NTZs in protecting highly mobile species, suggesting the need for larger or interconnected protected areas to accommodate such species' extensive range.

**Impact of UVC method**. Our findings indicate that higher logistical and financial investment does not necessarily yield more detection rates. Although the comb protocol demanded a greater sampling effort, we observed no marked improvement compared to the simpler cross protocol. These differences likely reflect variations in species behavior and habitat use, underscoring the need to account for species-specific traits when designing UVC surveys. Consequently, given the cross protocol's comparable—or even superior—performance at reduced cost and with fewer divers, we recommend it as a more efficient choice for long-term monitoring programs.

**Future research directions**. To enhance conservation efforts in MPAs, future research should focus on comprehensive monitoring across different habitats and depths. Key recommendations include:

- Monitoring of adjacent deeper sites: investigate vertical migrations and deeper habitat use of species like *D. dentex* and *S. umbra* to better understand their spatial ecology.
- Shallow coastal surveys for juvenile detection: adapt UVC protocols to include shallow coastal nursery sites to capture the complete life-history spectrum.

## Acknowledgments

We would like to thank the Natural Marine Park of Cape Corsica and Agriate (MNP CCA) for granting us the opportunity to conduct this study within the protected areas and for all the field assistance throughout the project. We also would like to thank the Regional Marine Fishers Committee of Corsica for granting the authorization to conduct the survey in the Nonza No Take Zone. Finally, we sincerely thank all the scientific scuba divers involved in the survey from the University of Corsica - Stella Mare marine platform and from the MNP CCA diving service.

## Author contributions

**Conceptualization:** Jessica Garcia, Christelle Paillon, Eric D. H. Durieux.

**Formal analysis:** Lucie Vanalderweireldt, Robin Bauknecht, Jessica Garcia, Manon Fournier, Antoine Brias.

**Funding acquisition:** Anthony Caro, Eric D. H. Durieux.

**Investigation:** Lucie Vanalderweireldt, Jessica Garcia, Christelle Paillon, Nicolas Tomasi, Anthony Caro, Eric D. H. Durieux.

**Methodology:** Lucie Vanalderweireldt, Christelle Paillon, Nicolas Tomasi, Eric D. H. Durieux.

**Project administration:** Eric D. H. Durieux.

**Resources:** Eric D. H. Durieux.

**Supervision:** Lucie Vanalderweireldt.

**Validation:** Lucie Vanalderweireldt, Eric D. H. Durieux.

**Visualization:** Lucie Vanalderweireldt.

**Writing – original draft:** Lucie Vanalderweireldt.

**Writing – review & editing:** Lucie Vanalderweireldt, Robin Bauknecht, Eric D. H. Durieux.

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
