## [Decision Letter · Decision Letter 0]

4 Feb 2025

PONE-D-24-44762Assessing the effectiveness of no-take zones on fish populations in the Marine Natural Park of Cap Corse and Agriate, Northwestern Mediterranean SeaPLOS ONE

Dear Dr. Vanalderweireldt,

Thank you for submitting your manuscript to PLOS ONE. After careful consideration, we feel that it has merit but does not fully meet PLOS ONE’s publication criteria as it currently stands. Therefore, we invite you to submit a revised version of the manuscript that addresses the points raised during the review process.

**Academic editor: In light of the reviewers' comments I suggest to adress their concerns. I agree with them that the manuscript need some revision to further improve it. I also suggest the authors to add some photos of the targeted fishes in the study along with some scheme of the method describe. I also suggest the authors to add, if possible, some raw footage of the methods as supplementary material.**

We look forward to receiving your revised manuscript.

Kind regards,

Alexandre Ribeiro da Silva

Academic Editor

PLOS ONE

Reviewers' comments:

Reviewer's Responses to Questions

**Comments to the Author**

1. Is the manuscript technically sound, and do the data support the conclusions?

Reviewer #1: No

Reviewer #2: Yes

2. Has the statistical analysis been performed appropriately and rigorously? 

Reviewer #1: No

Reviewer #2: No

3. Have the authors made all data underlying the findings in their manuscript fully available?

Reviewer #1: Yes

Reviewer #2: Yes

4. Is the manuscript presented in an intelligible fashion and written in standard English?

Reviewer #1: Yes

Reviewer #2: Yes

5. Review Comments to the Author

Reviewer #1: General view:

I have reviewed the manuscript titled "Assessing the effectiveness of no-take zones on fish populations in the Marine Natural Park of Cap Corse and Agriate, Northwestern Mediterranean Sea," submitted to PLOS One (PONE-D-24-44762). While the work is interesting, it does not significantly contribute new knowledge to the field of marine protected area (MPA) science. The paper's focus aligns with numerous studies conducted globally on MPA effects. Unfortunately, the manuscript fails to advance our understanding of MPAs by introducing novel methods or findings.

The authors concentrate on three apex predators as bioindicators in Mediterranean rocky shores. However, the Methodology section lacks a clear description of the sampling design, raising questions about the study's approach. The division into three sectors (Agriate, Canelle, and Nonza), with varying numbers of sampling points per sector, necessitates further explanation. A detailed description of the sampling protocol would enhance the article's clarity and understanding.

The explanation of the Underwater Visual Census (UVC) methodology is insufficient and difficult to comprehend. The use of two distinct protocols, with varying numbers of observers, raises concerns about potential biases in data collection. While the intent to employ diverse methods is understandable, the unequal sampling effort across methodologies and sectors compromises the data's credibility and overall interpretation.

I recommend a thorough revision of the manuscript, including a reanalysis of the data, before considering it for publication. In its current form, the article is primarily of local or regional interest and may be more suitable for a journal with a narrower geographic focus.

Specific comments:

Line 25 -> While the authors cite generalist papers, a relatively large number of studies have specifically examined MPA effects on these species: Harmelin-Vivien et al. 2008; Hackradt et al. 2014; Rojo et al. 2020, including outside the Mediterranean (Anderson et al. 2014 for E. marginatus).

Line 27 -> Similarly

Line 28 -> It is true that "assessing the population status of these three bioindicator species provides valuable ecosystem insights," but the manuscript should address new information on this issue.

Line 35 -> please provide reference for this citation

Line 43 -> This is not entirely accurate when considering the impact of fisheries. Some studies have found that indirect human activities, such as recreational diving, may not significantly impact fish populations, especially groupers (Pereñíguez et al. 2023). This statement requires rephrasing.

Line 52 -> However, the specific UVC method employed can directly influence the assessment of MPA effects on grouper, Dentex dentex and Sciaena umbra populations (see Rojo et al 2021).

Line 51 to 64 -> II suggest removing this paragraph. The discussion of UVC methodology is outdated and does not significantly enhance the manuscript. Instead, a focused discussion on the ecological context of S. umbra, E. marginatus, and D. dentex in France/Corsica would be more engaging for readers.

Line 70 to 77 -> This section appears to be more appropriate for the Materials and Methods section. Please reorganize the text accordingly.

Line 91 to 94 -> If Site A differs from the others, a clear justification for its inclusion is necessary. The rationale for its selection is unclear. Additionally, if evidence of differences between Site A and the NTZs exists, the authors should provide a convincing argument that these differences are primarily due to protection levels rather than habitat-mediated factors.

Line 137 -> The rationale for testing survey time, SST, and chlorophyll concentration is questionable. It is unlikely that top predators like groupers are significantly influenced by chlorophyll concentration, especially considering their relatively small home ranges.

Line 141 -> This corroborates my concern that the environmental data resolution the authors applied is biger tthan the fine-scale effects measurable by UVC. Including these variables in the GLM may introduce unnecessary variability, making it difficult to interpret the results. The authors may have overfitted the model by including excessive variables to explain the observed patterns, potentially leading to spurious correlations.

Line 171 -> This analysis is rather weak. A mixed-effects model, which can handle unbalanced designs and non-normal data, would be a more suitable approach. GLMMs with Gaussian distributions can correct for the bias introduced by the unbalanced design. Alternatively, PERMANOVA could be considered, as it is more robust to non-normal data and reduces the risk of Type II errors. Additionally, analyzing biomass, which incorporates both size and abundance, could provide a more comprehensive assessment.

Line 228 to 234 -> iI am not convinced by the proposed relationship between SST and chlorophyll concentration are real. This may be an artifact of the analysis. While the data suggests a correlation, a clear biological explanation for this relationship is lacking.

Line 238 → Authors did not find significant differences between protected and unprotected zones, and there are no visual indications of such differences. The text should avoid forcing the reader to perceive differences that do not exist.

Line 246 to 252-> This is not entirely new information. The influence of UVC protocols on data collection is well-established. The authors need to provide a clearer explanation of their data collection methods to address potential biases.

Discussion

The discussion is overly lengthy and repetitive, delving into extensive literature reviews and explanations that are well-established in MPA science. Notably, the authors have omitted citing relevant studies that examined similar questions using the same species and yielded similar results. It remains unclear whether this oversight stems from a lack of thorough literature review or a deliberate decision to exclude these studies. I recommend restructuring the discussion to emphasize novel findings and comparisons with existing literature, providing a concise and clear overview. Furthermore, I suggest the authors incorporate recommendations to enhance the management of the studied MPA or to improve national MPA management efficiency as a way to integrate the new insights presented in the article.

Reviewer #2: This study assessed the effectiveness of a no-take zone compared to two fished sites on Corsica. NTZ effectiveness was determined through UVCs of three predatory and threatened fish species. The study benefits from a relatively long temporal dataset, considerable diver time in conducting UVCs using two methods and across seasons and time of day, and good study design. The manuscript is very well written and effectively communicates the points of the authors.

However, reviewing the manuscript left me with several unanswered questions. This manuscript could either benefit from including those explanations in the text and/or as a supplement or the authors could choose to address these questions directly to the reviewers.

1. Inclusion of environmental factors: Inclusion of environmental factors in the GLMs seems to have been an afterthought. I greatly appreciate the detailed explanation in the Discussion about the effect of the significant environmental factors on the population densities and distribution of the three species. But the Introduction does not provide adequate context as to why they were included in the first place. The authors could consider including a few sentences in the Introduction explaining why certain environmental factors could influence fish population distributions. The inclusion of only significant environmental factors could, in a certain light, be viewed as cherry-picking or p-hacking. I encourage the authors to consider mentioning which other environmental factors were also analyzed and proved to be non-significant.

2. Study site effects vs fished and NTZ effects: The authors describe how one of the fished sites (Agriate) has a different habitat and bottom topography. Moreover, considerations of species size distributions are based upon site. Yet, in their presentation of the results the authors only present the effects of fished vs NTZ effects. I am curious as to what were the effects of individual sites on the GLMs? The manuscript would benefit greatly from a mention of whether this was tested and what the result was.

3. Study Protocol effects: Were divers in the Cross protocol trained using fish silhouette sizes before each season? If yes, please include a mention of this either by repeating the information or restructuring your description of the UVC protocols. Given the larger number of divers in the comb protocol vs cross protocol, did the authors consider the biases in fish size estimates and counts from differences in the number of observers? I appreciate the discussion of the pros and cons between these two methods. But the manuscript could benefit from a little more reflection on this or even potentially anecdotal mentions of inter-rater reliability if it was recorded during each season’s training session. Additionally, the authors could also reflect on the cost effectiveness of the cross vs comb technique, describing the differences in the amount of diver time as well as the area covered.

I commend the authors for their inclusion of descriptions of behaviours (eg. spawning aggregations, site fidelity, and nocturnal activity) affecting the distribution of fish species. The recommendations of adapting specific monitoring protocols to better estimate juveniles of populations or behavioral differences are also much appreciated.

Minor comments:

Page 2, line 45: Consider replacing “fishing cantonnement” with “fishing area” to make it readable for a wider audience.

Figures 3 and 4: Consider changing the color of the Cross Protocol bars and legend so that the lower limit of the black Standard Error bar is more visible.

6. PLOS authors have the option to publish the peer review history of their article (what does this mean?). If published, this will include your full peer review and any attached files.

Reviewer #1: No

Reviewer #2: **Yes: **Sahir Advani

---

## [Author Response · Author response to Decision Letter 1]

25 Apr 2025

All point-by-point responses to the editor’s and reviewers’ comments are provided in the attached PDF rebuttal letter.

---

## [Decision Letter · Decision Letter 1]

19 Jun 2025

Assessing the effectiveness of no-take zones on fish populations in the Marine Natural Park of Cap Corse and Agriate, Northwestern Mediterranean Sea

PONE-D-24-44762R1

Dear Dr. Vanalderweireldt,

We’re pleased to inform you that your manuscript has been judged scientifically suitable for publication and will be formally accepted for publication once it meets all outstanding technical requirements.

Kind regards,

Alexandre Ribeiro da Silva

Academic Editor

PLOS ONE

Additional Editor Comments (optional):

Reviewers' comments:

Reviewer's Responses to Questions

**Comments to the Author**

1. If the authors have adequately addressed your comments raised in a previous round of review and you feel that this manuscript is now acceptable for publication, you may indicate that here to bypass the “Comments to the Author” section, enter your conflict of interest statement in the “Confidential to Editor” section, and submit your "Accept" recommendation.

Reviewer #2: All comments have been addressed

Reviewer #3: All comments have been addressed

2. Is the manuscript technically sound, and do the data support the conclusions?

Reviewer #2: Yes

Reviewer #3: Yes

3. Has the statistical analysis been performed appropriately and rigorously? 

Reviewer #2: Yes

Reviewer #3: N/A

4. Have the authors made all data underlying the findings in their manuscript fully available?

Reviewer #2: Yes

Reviewer #3: Yes

5. Is the manuscript presented in an intelligible fashion and written in standard English?

Reviewer #2: Yes

Reviewer #3: Yes

6. Review Comments to the Author

Reviewer #2: The authors have done a commendable job in addressing both reviewer's comments and the manuscript has significantly improved as a result. I also appreciated the author's justification of the value of this locale-specific research - peer-reviewed science should not simply be published for novelty but also to help establish baselines and identify effective methodologies. I confidently recommend this manuscript for publishing and congratulate the authors and the scientific diving team for their considerable effort in undertaking this research.

Reviewer #3: I had the opportunity to review the manuscript titled "Assessing the effectiveness of no-take zones on fish populations in the Marine Natural Park of Cap Corse and Agriate, Northwestern Mediterranean Sea," submitted to PLOS ONE (PONE-D-24-44762R). Although the study has a regional scope, I believe it provides a valuable contribution to marine conservation by offering data that can support both local management actions and future research on the effectiveness of Marine Protected Areas (MPAs) in conserving fish populations. The manuscript is well-structured, with clear objectives, appropriate methodology, and a solid discussion. Therefore, I recommend its acceptance and wish the authors success with the publication.

7. PLOS authors have the option to publish the peer review history of their article (what does this mean?). If published, this will include your full peer review and any attached files.

Reviewer #2: **Yes: **Sahir Advani

Reviewer #3: No

---

## [Editor Report · Acceptance letter]

PONE-D-24-44762R1

PLOS ONE

Dear Dr. Vanalderweireldt,

I'm pleased to inform you that your manuscript has been deemed suitable for publication in PLOS ONE. Congratulations! Your manuscript is now being handed over to our production team.

Kind regards,

on behalf of

Dr. Alexandre Ribeiro da Silva

Academic Editor

PLOS ONE